# The Association of Vaccination for Common Adult Infectious Diseases and Uptake of COVID-19 Vaccines among 5,006,851 Veterans, 20 December 2020–31 October 2021

**DOI:** 10.3390/vaccines12020145

**Published:** 2024-01-30

**Authors:** Brady W. Bennett, Lawrence S. Phillips, Julie A. Gazmararian

**Affiliations:** 1Department of Epidemiology, Rollins School of Public Health, Emory University, Atlanta, GA 30322, USA; jagazma@emory.edu; 2Atlanta VA Medical Center, Decatur, GA 30033, USA; medlsp@emory.edu; 3Division of Endocrinology and Metabolism, Department of Medicine, School of Medicine, Emory University, Atlanta, GA 30322, USA

**Keywords:** COVID-19, vaccination, veterans, SARS-CoV-2, vaccine uptake

## Abstract

Disparities in vaccination coverage for coronavirus disease 2019 (COVID-19) in the United States (U.S.) are consistent barriers limiting our ability to control the spread of disease, particularly those by age and race/ethnicity. This study examines the association between previous vaccination for common adult infectious diseases and vaccination for SARS-CoV-2 among a cohort of veterans in the U.S. Sociodemographic and clinical data were utilized from three databases within the Veterans Health Administration included in the electronic health record. We examined the association of previous vaccination for common adult vaccinations through six separate multivariable logistic regression analyses, one for each previous vaccine exposure, adjusting for demographic and clinical variables. We also examined the association of receiving any one of the six common adult vaccinations and vaccination against SARS-CoV-2. Adjusted models indicate higher odds of vaccination for SARS-CoV-2 among those who received each of the previous vaccinations. Significant differences were also noted by race/ethnicity and age. Veterans who recorded receiving any one of the previous vaccinations for common adult infections had significantly greater odds of receiving any vaccination against SARS-CoV-2. Understanding veterans’ previous vaccination status can assist researchers and clinicians in impacting the uptake of novel vaccines, such as vaccination against SARS-CoV-2.

## 1. Introduction

Disparities in vaccination coverage for coronavirus disease 2019 (COVID-19) in the United States (U.S.) are consistent barriers limiting our ability to control the spread of disease, particularly those by age and race/ethnicity. In the initial months of the COVID-19 pandemic, data from the Centers for Disease Control and Prevention (CDC) indicate that Black and Hispanic persons in the U.S. were more likely to test positive or experience severe morbidity and mortality than White, non-Hispanic persons [1]. However, early evidence from the U.S. Veterans Affairs Health System (VHA), the largest integrated healthcare system in the U.S., did not show similar disparities in mortality [2].

The VHA system in the United States serves approximately 9 million veterans each year in over 1300 health care facilities including 172 medical centers and 1138 outpatient clinics [3]. The Veteran’s Administration (VA) Information Resource Center (VIReC), which exists to advance VA research and data management, fosters data use for research and quality improvement across numerous internal and external datasets [4,5]. Data available for researchers include patient demographics, healthcare encounters, performance measures, financials, and pertinent data from the Centers for Medicare and Medicaid Services (CMS) and the Department of Defense (DOD), amongst others [5]. These data are integrated into several data sources for cross-disciplinary research to both internal and external VA researchers.

The VHA patient population is a subset of the overall veterans requiring active-duty service without a dishonorable discharge. Additionally, VHA patients are more likely to have a service-connected disability and must meet income threshold requirements [6,7]. Veterans, and especially those accessing the VHA, are at increased risk for severe outcomes and experience a greater disease burden from COVID-19 [8,9] compared to the general population. Therefore, veterans are a population that could substantially benefit from the utilization of severe disease prevention mechanisms such as vaccination.

In August 2020, anticipating approval of a vaccine for SARS-CoV-2, the virus that causes COVID-19, the VHA set up a Vaccine Integrated Project Team composed of six sub-groups incorporating input from the National Center for Ethics in Healthcare to determine the proper risk stratification for distribution among veterans [10]. When the Food and Drug Administration (FDA) approved an emergency use authorization (EUA) for the Pfizer/BioNTech vaccine on 11 December 2020, the Department of Veteran’s Affairs (VA) became a primary conduit of vaccination against SARS-CoV-2.

By 7 March 2021, less than 3 months after the initial EUA was approved by the FDA, 23% of the general VHA population had received at least one vaccine injection with 94% vaccine efficacy against infection amongst those who received an initial primary series [11]. Although reports through the summer of 2021 indicated unequal uptake of vaccination across military branches and hesitancy among VHA employees and veterans similar to those among U.S. adults [12,13,14,15,16,17,18], disparities in vaccine uptake seen in the general U.S. population [1] were mitigated in the VHA population [19]. Veterans historically have a higher rate of vaccination compared to the general population [20,21]; however, they often still fall well below national guidelines [21]. From 2016 to 2018, only 45% of male veterans aged 25–64 years had received the influenza vaccine in the previous year [22], slightly higher than that of U.S. adults ≥ 18 years old who were vaccinated in the same time period [23] but well below the Healthy People 2030 immunization goal [24]. Similar trends are seen in other common adult vaccinations such as pneumococcal [22] and zoster/shingles [25]. With each novel variant of SARS-CoV-2 (and each iteration of the COVID-19 vaccine), it is imperative to understand how vaccination and booster uptake can be improved and what factors, in addition to known sociodemographic factors [26], are associated with vaccination. Therefore, trends of uptake among other common vaccinations may provide insight into vaccine uptake, or lack thereof, for COVID-19 and help guide practitioners, health educators, and vaccine promotion campaigns to improve vaccine coverage among veterans.

This goal of this study is to determine whether previous vaccination against common adult infectious diseases, analyzed separately, is associated with a higher likelihood of vaccination against SARS-CoV-2. Additionally, differences in the likelihood of vaccination for COVID-19 across demographic characteristics, namely race/ethnicity and age, are examined. Considering the influence of demographic characteristics is critical to fully understanding where vaccine uptake is lagging and where valuable resources should be directed.

## 2. Materials and Methods

### 2.1. Study Design and Study Population

Utilizing a single, national electronic health record system and two socio-demographic databases, a cross-sectional study of veterans in active care at the Veterans Health Administration (VHA) was conducted. The study population included all veterans with at least one inpatient hospital stay or outpatient encounter between 1 January 2018, and 30 October 2021 (Figure 1). The analytic dataset included those who had a record of COVID-19 vaccination status in the VHA electronic health record from 15 December 2020, to 30 October 2021, and had no missing data for all other variables of interest.

The study was exempted by the Emory University Institutional Review Board and the Atlanta VA Research and Development Committee. It is considered Health Insurance Portability and Accountability Act (HIPAA) compliant.

### 2.2. Data Sources

Three primary data sources that make up the VHA EHR were used for this study: (1) Corporate Data Warehouse (CDW), a relational database aggregating patient data from all VHA facilities from 1999 to present, accessed from the VA Informatics and Computing Infrastructure (VINCI); (2) the United States Veterans Eligibility Trends and Statistics (USVETS) database, an integrated dataset of veteran demographic and socioeconomic data; and (3) the VHA COVID-19 shared data resource for data related to SARS-CoV-2 vaccination.

### 2.3. Outcome

The outcome of interest was vaccination for SARS-CoV-2 (COVID-19) and was categorized as “any vaccination” compared to “no vaccination” within the VA system.

### 2.4. Exposures

Six exposure variables were considered, representing prior vaccination for influenza; pneumonia; tetanus-diphtheria; varicella; measles, mumps, and rubella (MMR); and meningitis. Each vaccine was considered separately as a primary exposure and categorized as “ever vaccinated” or “not vaccinated”.

### 2.5. Covariates

Covariates of interest included COVID-19 test positivity, gender, race/ethnicity, rurality, marital status, household size, household income, age and Elixhauser comorbidity index. Test positivity was operationalized as “ever positive” and “never positive”. Self-reported gender was categorized as either “male” or “female” (there is no distinction available for transgender men, transgender women, non-binary persons, or other identifications), and race/ethnicity were characterized as “White, non-Hispanic”, “Black, non-Hispanic”, “Hispanic”, and “Other”, where “Other” represented persons who identified as Asian, Native Hawaiian/Pacific Islander, or American Indian/Alaska Native. Rurality utilizes the rurality-urban commuting area (RUCA) codes from the U.S. Department of Agriculture (USDA) to categorize whether a patient lives in a rural, urban, or unknown rurality area. RUCA codes classify census tracts using measures of population density, urbanization, and daily commuting [27]. Marital status was categorized as “married” or “not married”. Household size was categorized as having 1, 2, 3, or ≥4 persons in the household. Household income was categorized as “<USD 40,000”, “USD 40,000-USD 74,999”, and “>USD 75,000”. Age was dichotomized to represent age <65 years vs. ≥65 years. The Elixhauser comorbidity index, a method of categorizing comorbidities of patients based upon International Classification of Diseases (ICD) diagnosis codes [28], is continuous. It consists of 31 comorbidities shown to be associated with increases in length of hospital stay, hospital charges, and mortality and includes mental health disorders, drug and alcohol abuse, coagulopathy, and fluid and electrolyte disorders amongst others [29,30].

### 2.6. Analysis

Cross tabulations of COVID-19 vaccination status by both primary exposures and covariates were calculated, and a chi-square test was used to determine associations. Multivariable logistic regression models were examined with each prior infectious disease vaccination as an exposure. We also considered whether there were substantial differences in the effect of our exposures on the outcome if individual chronic conditions were included as covariates rather than the Elixhauser comorbidity index. The comorbidities include cancer, alcohol, asthma, chronic obstructive pulmonary disease (COPD), coronary artery disease (CAD), chronic kidney disease (CKD), diabetes, heart failure, human immunodeficiency virus (HIV), hypertension, liver disease, obesity, stroke, and tobacco use. Ultimately, there was no noticeable difference in odds ratios when utilizing separate variables for individual chronic conditions compared to the Elixhauser comorbidity index. Therefore, the Elixhauser comorbidity index was used in all models rather than the individual comorbidities.

Models also explored potential effect measure modification by age, race/ethnicity, and gender. Stratified models were utilized to determine whether the effect of each exposure variable on the outcome varied by age and race/ethnicity. Model results are presented as stratified by binary age (≥65 years vs. <65 years) and race/ethnicity (non-Hispanic White, non-Hispanic Black, Hispanic, Other) to account for this effect measure modification.

A final model was completed to determine whether having any one of the six routine vaccines predict vaccination for COVID-19. COVID-19 test positivity, gender, race, rurality, marital status, household size, household income, Elixhauser comorbidity index, and age were included as covariates.

## 3. Results

Between February 2020 and October 2021, 8,059,078 veterans had a COVID-19 vaccine status defined as vaccinated or never vaccinated. Of those, nearly 70% of persons who had never tested positive for COVID-19 had received at least one COVID-19 vaccination. Conversely, nearly 55% of those who had at least one positive test received a COVID-19 vaccination (Table 1). Furthermore, approximately 48% of both men and women had received at least one vaccination shot, while persons age ≥65 years had a higher percentage vaccinated (51.7%) compared to those <65 years old (43.3%). Individuals who identified as Black, non-Hispanic had a higher vaccination rate (54.7%) than those who identified as not Black, non-Hispanic (47.8%), and individuals who were married had a slightly higher vaccination percentage (49.9%) compared to those who were not married (48.0%). Persons who live in an urban setting had a slightly higher rate of vaccination (49.6%) compared to those who live in a rural setting (44.7%) or unknown (42.2%). There was a larger percentage of persons vaccinated for COVID-19 as household size increased (48.0–50.6%) and as household income increased (46.1–51.0%). Lastly, there was a slightly lower mean Elixhuaser comorbidity score for persons who were vaccinated (4.86) compared to those who were not vaccinated (4.91).

Cross-tabulations of six common adult immunizations with current COVID-19 vaccination status indicated higher vaccination for COVID-19 among those who have previous immunization for influenza (55.2%), pneumonia (58.3%), tetanus-diphtheria (53.8%), varicella (68.8%), meningococcal (59.6%), and MMR (52.7%) (all *p* < 0.0001) compared to those who do not have a record of a previous immunization (Figure 2).

Unadjusted models indicated increased odds of vaccination for COVID-19 among persons with previous adult vaccinations compared to those without previous adult vaccinations (Table 2). Effect measures ranged from 1.22 (95% CI = 1.21, 1.24) higher odds (previous MMR vaccination) to 3.47 (95% CI = 3.46, 3.48) higher odds (previous varicella vaccination). Persons who received any previous vaccine had nearly four-fold higher odds of receiving a vaccination for COVID-19 compared to those who received no previous vaccination in crude models (OR = 3.76, 95% CI = 3.74, 3.77).

Adjusted models indicated varying degrees of association with COVID-19 vaccination across different vaccine types. Persons who received a previous vaccination for influenza had 2.86 (95% CI = 2.85, 2.86) times the odds of receiving at least one COVID-19 vaccination shot compared to those who did not receive a previous vaccination for influenza. Likewise, those who received a previous vaccination for pneumonia had 2.51 (95% CI = 2.48, 2.53) times the odds; those who received vaccination for tetanus-diphtheria had 1.78 (95% CI = 1.75, 1.81) times the odds; those who received vaccination for varicella had 2.67 (95% CI = 2.65, 2.69) times the odds; those who received previous vaccination for MMR had 1.15 (95% CI = 1.07, 1.24) times the odds; and those who had previous vaccination for meningitis had 1.53 (95% CI = 1.43, 1.62) times the odds of receiving at least one COVID-19 vaccination compared to those who were not previously vaccinated. Persons who received any one of the previous vaccination exposures had 3.54 (95% CI = 3.46, 3.61) times the odds of receiving at least one COVID-19 vaccination shot compared to those who received no previous vaccinations.

Stratified models indicated substantial differences in the odds of at least one COVID-19 vaccination for those who received a previous vaccination compared to those who did not by race/ethnicity and age (Table 2). For all previous vaccination exposures, apart from previous MMR vaccination, we found the effect of previous vaccination to be in the same direction regardless of race/ethnicity (i.e., persons who received a previous vaccination had increased odds of receiving at least one COVID-19 vaccination shot compared to those who did not receive a previous vaccination regardless of race/ethnicity). For example, the odds of at least one COVID-19 vaccination among those who received a previous influenza vaccination was 2.46 (95 CI = 2.41, 2.51) for Black individuals, 3.19 (95% CI = 3.15, 3.23) for White individuals, 2.71 (95% CI = 2.56, 2.84) for Hispanic individuals, and 2.75 (95% CI = 2.51, 3.01) for all others, compared to those who did not receive a previous influenza vaccination. However, for those who received a previous MMR vaccination, Black individuals had 0.87 (95% CI = 0.81, 0.94) times the odds, and Hispanic individuals 0.81 (95% CI = 0.72, 0.90) times the odds of receiving at least one COVID-19 vaccination shot compared to those who did not receive a previous MMR vaccination, whereas White individuals who received a previous MMR vaccination had 1.08 (95% CI = 1.04, 1.13) times the odds of receiving at least one COVID-19 vaccination shot.

Additionally, the magnitude of the effect differed by race/ethnicity across previous vaccination exposures. Black and Hispanic individuals who had previously received immunization for influenza, pneumonia, tetanus-diphtheria, MMR, and meningitis had consistently lower odds of subsequent vaccination for COVID-19 compared to White individuals who had received the same previous immunizations; however, Black individuals who had a previous varicella immunization had roughly the same odds of vaccination for COVID-19 as White individuals who received previous immunization for varicella. Persons categorized as races other than Black, White, or Hispanic (those who identify as Asian, Native American/Alaska Native, Hawaiian/Pacific Islander, or multiracial) had no consistent trend in comparison to other demographic groups. Persons categorized as Other with a previous influenza immunization had greater odds of vaccination for COVID-19 compared to Black or Hispanic persons, but lower odds compared to White. However, for persons with a previous tetanus-diphtheria or MMR immunization, persons who are categorized as Other had higher odds of vaccination for COVID-19 compared to Black, White, or Hispanic persons.

We also found differential effect size by age, though the direction of effect remained consistent. Persons ≥65 years of age who had a previous immunization for influenza, pneumonia, tetanus-diphtheria, MMR, or meningitis had higher odds of vaccination for COVID-19 compared to those who did not have a previous vaccination than among persons <65 years of age. However, among persons <65 years of age who received a previous varicella immunization, the odds of at least one vaccination for COVID-19 was greater than the corresponding odds among persons ≥65 years of age. Additionally, persons <65 years of age who received a previous vaccination for MMR showed no significantly different odds of receiving at least one COVID-19 vaccination shot compared to those who did not receive a previous MMR vaccination (OR = 0.99, 95% CI = 0.95, 1.03). Persons ≥65 years of age who received any previous vaccine also had higher odds of receiving a vaccination for COVID-19 compared to persons <65.

## 4. Discussion

Our findings indicate that veterans who have received routine vaccination for common infectious diseases were more likely to receive at least one COVID-19 vaccination shot. While the utility of previous influenza vaccination as a predictor of COVID-19 vaccination has been researched [31], this study is the first of our knowledge to expand to other routine vaccination exposures. By examining the association between previous vaccination for additional infectious diseases and vaccination for COVID-19, future vaccination campaigns can be targeted to increase uptake, provide education, and, ultimately, decrease infection rates among veterans in the United States.

This study builds on the findings of other recent studies among those receiving care in the VHA that examined differences in vaccination by race. Their results found that Black, Hispanic, and Asian American/Pacific Islander individuals were more likely to receive a COVID-19 vaccination than White individuals [19]. Our findings examined a different aspect of this relationship by considering whether race/ethnicity (and age) modified our primary exposure-outcome association. While the other study’s authors concluded differences in vaccination uptake by race/ethnicity, we found that receiving previous routine vaccinations were associated with increased odds of vaccination for COVID-19 regardless of race/ethnicity; however, the magnitude of that association differed by race/ethnicity and by age.

Our findings also confirm results from a previous study that found influenza vaccination to be a predictor of vaccination for COVID-19 among veterans [32]. Our results also indicate increased likelihood of vaccination for COVID-19 among those who received vaccination for influenza while also finding increased COVID-19 vaccination among those with other routine vaccinations.

The results of this study also indicate that there is increased likelihood of COVID-19 vaccination if receiving vaccines across the lifespan. For example, we found increased likelihood of COVID-19 vaccination for persons who had received a vaccine for MMR, tetanus-diphtheria, or meningitis. These are typically given in childhood or early adulthood (e.g., college entry) and are each required for military service. However, they may also be renewed in 10+ year increments throughout one’s life as immunity decreases. In our dataset, the prevalence of both MMR and meningitis vaccinations are <1% (Appendix B), indicating that even if with smaller percentages of ongoing vaccination, previous vaccination for MMR and meningitis remain predictive of vaccination for COVID-19.

Additionally, our findings demonstrating differences in vaccination rates by race and by age are consistent with results in both the veteran and general population [33,34,35,36]. Moreover, we identified that Black veterans who had received a previous vaccination were up to 29% less likely to obtain vaccination for COVID-19, and Hispanic veterans were up to 18% less likely to obtain vaccination for COVID-19 compared to their White counterparts. Furthermore, veterans that are ≥65 years old are also up to 51% more likely to become vaccinated against COVID-19 if they received a previous vaccination compared to those <65 years old. These results represent substantial differences in vaccine uptake by race and age and contribute to the literature on disparities in vaccine uptake both within and external to the VHA by specifically considering vaccination rates by race and age for those who have received other common adult vaccinations.

There are several possible explanations for our findings. First, the VHA, serving a veteran population with required vaccination for many infectious diseases for service, including five of the six routine vaccines included in this study, may be more accepting of vaccination than other populations, though hesitancy does exist [37,38]. Additionally, improved access to care among VHA-utilizing veterans compared to the general public, and even non-VHA-utilizing veterans, may also contribute to increased uptake and high odds of vaccination in general, not just for COVID-19 [21].

### Policy/Program Implications

Finally, there are several policy and programmatic implications from our analysis. Our general finding that documented uptake of routine vaccinations is associated with increased likelihood of vaccination for COVID-19 indicates that those who are not utilizing routine vaccinations may have underlying fears, concerns, and/or hesitancies around vaccination more broadly. This requires targeted education initiatives to dispel misinformation about vaccinations, clear communication about the safety and benefits of vaccines, and more consistent outreach to groups with lower general uptake of vaccines. Additionally, our finding that, among those who are utilizing routine vaccinations against common infectious diseases, differences exists by race/ethnicity and by age indicates that additional outreach and effort must be applied to ensure equitable availability of vaccines as well as uptake.

It is also imperative to consider how these results may be interpreted in the context of ongoing efforts to vaccinate for COVID-19. Currently, the majority of individuals in the United States are eligible for approximately annual vaccination for COVID-19, similar to influenza vaccination. Our study elucidates key characteristics (i.e., previous vaccine uptake for routine vaccinations) that predict likelihood of initial COVID-19 vaccine uptake and differences by age and race/ethnicity. This provides necessary contextual information for providers, vaccine program personnel, and researchers seeking to understand opportunities for improved ongoing COVID-19 vaccine uptake.

## 5. Strengths

There are at least three key strengths to this study. First, the utilization of the Veterans Health Administration system eliminates a key barrier to care that most of the non-veteran U.S. population has—differential access to care by income and insurance status. All eligible veterans have access to the VHA system with comprehensive, universal medical care—creating a unique population for research in the United States [7]. Second, this study used a large, robust sample encompassing over 5,000,000 individuals, which allowed us to consider modifications in vaccine uptake by age and race. Third, to our knowledge, this is the first analysis to consider other routine vaccinations, other than influenza, as a predictor of COVID-19 vaccination. Extending our understanding of the predictors and characteristics of individuals who do and those who do not utilize new and important vaccines is critical to our ability to ensure equitable and widespread uptake.

## 6. Limitations

Despite these strengths, there are several limitations to this study. First, there were two variables included in the regression models that contain imputed data, household income and marital status, which introduce the opportunity for misclassification. These data were imputed by a third party utilizing a proprietary algorithm whose method was not available to the researchers. However, additional sensitivity analyses were conducted, indicating that there were not substantial differences in the association of our primary exposures when using the non-imputed data compared to the imputed. Not all covariate data in the USVETS data system were imputed, and therefore, some missingness still existed in the analytic dataset. Second, we did not examine the odds of receiving 1, 2, or 3 injections of vaccine for COVID-19. We were interested in higher-level associations with COVID-19 vaccinations rather than shot-specific implications. Therefore, we did not assess patients’ completion of a primary vaccination series versus less than a complete primary vaccination series or the association with specific vaccine types (i.e., Pfizer vs. Moderna vs. Janssen), which may also be affected by access at certain points in time. Additionally, the vaccination status represented in this analytic dataset may not include those who received vaccinations outside of the VHA, leading to misclassification of vaccination status. However, we do not believe that this occurred differentially by race or age. Lastly, there were approximately 3 million individuals removed from our dataset due to missing data. Although, our sample size remained substantially large for generalizations representing over half of veterans seeking care at the VHA, it is possible that the excluded individuals are categorically different than those who were included, creating bias in our analysis. Future analyses should consider whether previous vaccination against common infectious diseases predicts the number of doses received. This would provide researchers a better understanding of COVID-19 booster uptake and ongoing vaccination and the characteristics of those most likely to utilize subsequent vaccinations. Additionally, further inquiries are needed that examine the socio-political nature and disparities of vaccine uptake in the veteran community.

## 7. Conclusions

With the recent approval and roll-out of a bivalent vaccine incorporating more recent variants of COVID-19, it is imperative to continue and improve vaccination, especially among high-risk populations, such as veterans. Improving vaccination rates for COVID-19 will help mitigate the health effects of COVID-19 infection, particularly for those at highest risk of severe disease and death. With each subsequent variant and lagging vaccination rates nationally and among veterans, it is vital to understand factors that are associated with higher vaccination rates for COVID-19 and disparities that may exist between groups. Most importantly, addressing these disparities in vaccine uptake through education, dispelling of misinformation, and ensuring access to care are critical to preventing infection and slowing the spread of COVID-19 and other adult infections.

## Figures and Tables

**Figure 1 vaccines-12-00145-f001:**
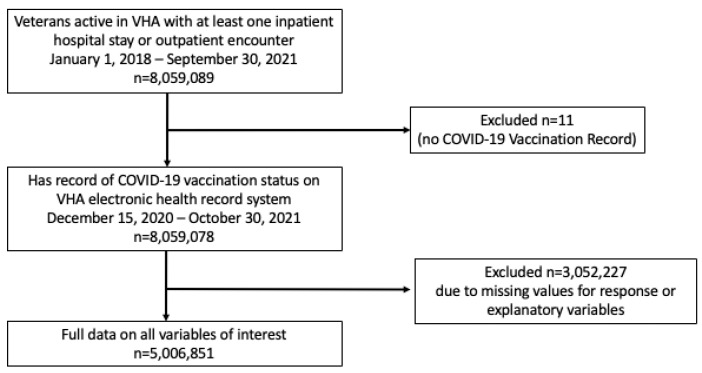
Study population—veterans utilizing VHA facilities between 15 December 2020, and 30 October 2021.

**Figure 2 vaccines-12-00145-f002:**
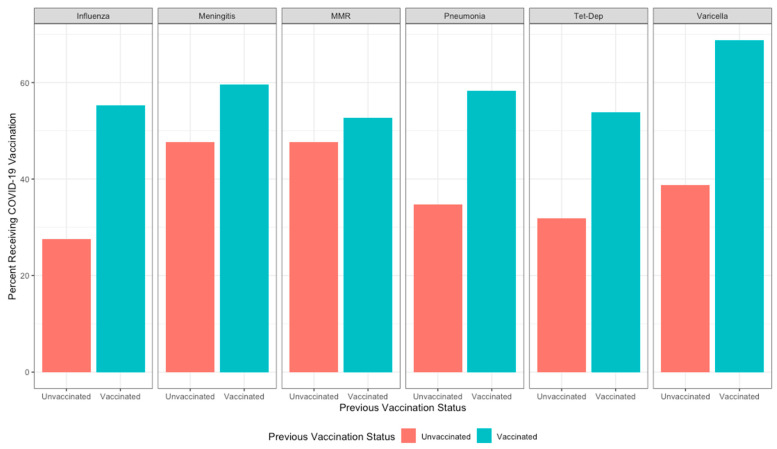
Common adult vaccination rates by COVID-19 vaccination status among veterans receiving care at Veterans Health Administration facilities, 15 December 2020–30 October 2021.

**Table 1 vaccines-12-00145-t001:** Demographic characteristics of COVID-19 vaccine status among veterans in the United States, 15 December 2020–30 October 2021 (*n* = 5,006,851).

Characteristics	Unvaccinated (*n* = 4,217,710)	Vaccinated (*n* = 3,841,368)	*p*-Value
**COVID-19 Test Positivity (%)**			<0.001
Ever Yes	30.7	69.3	
Ever No	45.3	54.7	
**Gender (%)**			<0.001
Female	52.4	47.6	
Male	52.1	47.9	
**Age (%)**			<0.001
<65 years	56.7	43.3	
≥65 years	48.3	51.7	
**Race/Ethnicity (%)**			<0.001
White	52.8	47.2	
Black/African American	45.3	54.7	
Hispanic	31.3	68.7	
Other Races/Ethnicities	53.6	46.4	
**Marital Status (%)**			<0.001
Not Married	52.0	48.0	
Married	50.1	49.9	
**Rurality/Urbanicity (%)**			<0.001
Rural	55.3	44.7	
Urban	50.4	49.6	
Unknown	57.8	42.2	
**Household Size (%)**			<0.001
1 person	52.0	48.0	
2 people	49.9	50.1	
3 people	49.6	50.4	
4+ people	49.4	50.6	
**Household Income (%)**			<0.001
<USD 40,000	53.9	46.1	
USD 40,000-USD 74,999	50.8	49.2	
≥USD 75,000	49.0	51.0	
**Elixhauser score, mean**	4.91	4.86	<0.001

**Table 2 vaccines-12-00145-t002:** Odds of receiving COVID-19 vaccination among veterans by previous vaccination, stratified by race/ethnicity and age.

	Influenza	Pneumonia	Tetanus-Diphtheria	Varicella	MMR	Meningitis	Any Previous Vaccine
Crude OR	3.23 (3.22, 3.24)	2.62 (2.61, 2.63)	2.51 (2.50, 2.51)	3.47 (3.46, 3.48)	1.22 (1.21, 1.24)	1.62 (1.59, 1.65)	3.76 (3.74, 3.77)
Adjusted OR *	2.86 (2.85, 2.86)	2.51 (2.48, 2.53)	1.78 (1.75, 1.81)	2.67 (2.65, 2.69)	1.15 (1.07, 1.24)	1.53 (1.43, 1.62)	3.54 (3.46, 3.61)
Stratified OR by Race **							
Black	2.46 (2.41, 2.51)	2.36 (2.32, 2.41)	1.75 (1.71, 1.78)	2.79 (2.73, 2.85)	0.87 (0.81, 0.94)	1.54 (1.43, 1.66)	2.99 (2.8, 3.10)
White	3.19 (3.15, 3.23)	2.54 (2.51, 2.57)	1.81 (1.78, 1.83)	2.74 (2.71, 2.77)	1.08 (1.04, 1.13)	1.55 (1.47, 1.64)	3.96 (3.85, 4.07)
Hispanic	2.71 (2.56, 2.84)	2.21 (2.14, 2.29)	1.56 (1.49, 1.63)	2.72 (2.61, 2.83)	0.81 (0.72, 0.90)	1.46 (1.28, 1.68)	3.02 (2.76, 3.29)
Other	2.75 (2.51, 3.01)	2.26 (2.10, 2.43)	1.85 (1.69, 2.03)	2.53 (2.35, 2.72)	1.12 (0.91, 1.37)	1.35 (0.99, 1.84)	3.11 (2.63, 3.67)
Stratified OR by Age ***							
<65 years	2.67 (2.64, 2.72)	2.19 (2.16, 2.21)	1.62 (1.59, 1.64)	3.06 (3.02, 3.11)	0.99 (0.95, 1.03)	1.48 (1.41, 1.55)	2.83 (2.76, 2.91)
≥65 years	3.47 (3.41, 3.53)	3.12 (3.06, 3.18)	2.04 (2.00, 2.08)	2.47 (2.44, 2.50)	1.49 (1.32, 1.68)	1.74 (1.59, 1.89)	4.57 (4.40, 4.73)

* Model adjusts for “ever positive”, age, race, rurality, marital status, household size, household income, and Elixhauser score; ** race stratified model adjusts for all in adjusted model except race; *** age stratified model adjusts for all in adjusted model except age.

## Data Availability

The United States Department of Veterans Affairs (VA) places legal restrictions on access to veteran’s health care data. The data that support the findings for this study are not permitted to leave the VA firewall without a Data Use Agreement. VA data are made freely available to researchers behind the VA firewall with an approved VA study protocol. For more information, please visit https://www.virec.research.va.gov.

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
