# Peer review of "The Association of Vaccination for Common Adult Infectious Diseases and Uptake of COVID-19 Vaccines among 5,006,851 Veterans, 20 December 2020–31 October 2021"

_vaccines, 2024, doi:10.3390/vaccines12020145_

Round 1

Reviewer 1 Report

Comments and Suggestions for Authors

This study identifies how COVID-19 vaccine update can relate to previous vaccinations of six common vaccines. Receiving these vaccines was associated with increased odds of receiving COVID-19 vaccines, with differences in effect noted by race and age. The authors bring up the important point that promoting vaccination will require additional effort among those who had not received previous routine vaccinations in their lifetimes. The methods used are sound and support the conclusions made.

MAJOR COMMENTS

1. There seems to be a disconnect with the analyses described in the Methods section and the model results in the Results section. The Analysis subsection describes multivariable regressions followed by another set of multivariable regressions with interaction terms to evaluate effect measure modification. It then describes “a final model,” but it is unclear how this differs from the “final” models in the preceding paragraph.

The results section presents the results of the multivariable regressions, but then includes stratified regressions. Stratified analyses typically consist of separate analyses for each subgroup rather than interaction terms. And results for “a final model” are not included. The Analysis subsection would benefit from further clarity, and the analyses presented in the Methods and Results sections should match.

2. The last paragraphs of the Results section present analyses by race and age. Throughout the paragraphs, the odds of being vaccinated is compared across race and age groups. Are these referring to odds of being vaccinated differing by race/age or are these actually referring to odds ratios between previous vaccine exposure differing by race/age? The latter is what is presented in Table 3.

3. The Methods section and Figure 1 show that a nontrivial number of observations were removed due to missing covariate data. The limitations subsection, however, mentions the use of imputed data. What is the distinction between observations removed due to missing data and observations that include imputed missing data? Also, it is important to describe in the Methods section whether missing data were imputed and how missing data were imputed.

MINOR COMMENTS

4. All six previous routine vaccines were associated with greater likelihood of receiving a COVID-19 vaccine, though these all are typically given at different stages of life and with different frequencies. For example, MMRs are typically given in childhood, while influenza vaccines are given throughout one’s lifetime, with emphasis in later years. Depending on what information is available in the data set, it may be worth noting whether different vaccines typically were given in childhood/adult years. This is a potentially noteworthy finding if vaccines given in childhood are predictive of getting vaccinated as an adult.

5. It would be interesting to show what percentage of the study population had received each of the six routine vaccines as well as the percentage that had received any vaccine. This would show the relative sizes of the exposure groups and then show, through the percentage of the population having received any vaccine, whether there was substantial overlap (i.e. whether most people received either all or none of the vaccines).

6. Are all covariates present in Table 1 as of Oct 30, 2021? While some are clearly not time-varying, others, especially COVID-19 infection, are. While this does not invalidate any analyses or results, it is important context because it does mean that it is possible for first COVID-19 infections to have occurred after being vaccinated.

7. It would be helpful to contextualize the results in terms of ongoing efforts for COVID-19 vaccination. In 2024, it is assumed that ongoing vaccination will consist of annual doses, resembling annual influenza vaccination. While it is likely that the results of this study focusing on the initial doses will apply to ongoing annual vaccination, it would be beneficial to add to the discussion how this will likely apply.

Reviewer 2 Report

Comments and Suggestions for Authors

Comments

This study is very interesting and well-written. This study examined the association of vaccination for common adult infectious diseases and uptake of covid-19 vaccines among veterans in US. However, lacks some important information in the main text. Here are some points I would like the authors to consider to further highlight the contribution of the study.

Abstract:

1.In the abstract section, it is important to clearly articulate the methods and data sources used in the study.

Introduction:

1.In the introduction section, a brief overview of the U.S. Veterans Affairs Health System (VHA) database and system should be provided. This overview can include information about the VHA's role in providing healthcare to U.S. veterans, the scope and scale of the database, and how it is utilized for research and healthcare improvement. This context will help readers understand the relevance and significance of the VHA system in the context of the study.

Materials and Methods:

1.For Figure 1, careful editing and enhancement are required as there are currently errors present. For instance, a period is missing in the first box on the right side. Attention to such details is crucial for ensuring the clarity and professionalism of the figure.

2.To assist readers in understanding the baseline characteristics of the sample with missing values, a descriptive analysis of the over 3 million samples with missing data should be conducted. This analysis can include key demographic and clinical characteristics, such as age, gender, health conditions, and any other relevant variables. Presenting this information will help readers assess the potential impact of excluding these samples on the study's findings and validity.

3.Listing the covariates in a table format would indeed make the information clearer and more accessible. A well-organized table can effectively present the covariates, their definitions, and possibly their sources or categories, enhancing the readability and comprehension of the study.

Results

1.It is indeed important to specify the vaccination rates for the six common adult vaccinations in the study. The varying rates of these vaccinations, along with the differing sample sizes, can significantly influence the likelihood of individuals receiving the Covid vaccine. Clarifying these rates will provide valuable context for understanding the study's findings, particularly in relation to how prior vaccination behaviors may impact Covid vaccine uptake.

2.In the results section, it is crucial to meticulously check for issues related to grammar, spacing, and formatting. For instance, the presentation of 95% confidence intervals (CI) should be consistent throughout. Some are currently formatted as (95% CI = 0.72, 0.90), while others are presented as (1.04, 1.13). Uniformity in formatting these details is essential for maintaining clarity and professionalism in the document.

Round 2

Reviewer 1 Report

Comments and Suggestions for Authors

Thank you for revising the manuscript. All of my previous comments have been adequately addressed, and the current version's methods and interpretations are clearer.